**Subject Category:**
Biology (whole organism)

biomaterials/biomedical engineering

electrospinning, icariin, osteoarthritis, cartilage degeneration, subchondral bone

**Author for correspondence:**
Yang Wang
e-mail: wangyang99@jlu.edu.cn

# PLGA scaffold carrying icariin to inhibit the progression of osteoarthritis in rabbits

Chang Fu Zhao[1], Zhen Hua Li[2], Shao Jun Li[2], Jian An Li[3], Ting Ting Hou[4] and Yang Wang[1]

[1]Department of Orthopaedics, China-Japan Union Hospital, Jilin University, 126 Xiantai St, Changchun, People's Republic of China
[2]Department of Orthopaedics, The Affiliated Hospital to Changchun University of Traditional Chinese Medicine, 1478 Gongnong Road, Changchun, People's Republic of China
[3]Department of Orthopaedics, Tianjin Hospital, 406 Jiefang South Road, Tianjin, People's Republic of China
[4]Department of Orthopaedics, The Second Hospital of Jilin University, 218 Zi qiang Street, Changchun, People's Republic of China

 YW, 0000-0002-7251-3767

Icariin, the main effective component extracted from epimedium, has been shown to stimulate osteogenic differentiation and bone formation and to increase synthesis of the cartilage extracellular matrix. However, there has been little study on the effects of icariin on osteoarthritis. In this study, we loaded icariin onto poly(lactic-co-glycolic acid) (PLGA) electrospinning. The aim of this study was to explore a composite scaffold and to inhibit the progression of osteoarthritis. Our main experimental results demonstrated that the PLGA/icariin composite spinning scaffold had higher hydrophilicity, and icariin was released slowly and steadily from the scaffold. According to the results of an MTT test, immunofluorescence staining, an alkaline phosphate activating assay and a real-time polymerase chain reaction (RT-PCR) assay, the PLGA/icariin composite scaffold had good biocompatibility. In models of osteoarthritis, the results of a RT-PCR assay indicated that the PLGA/icariin scaffold promoted the synthesis of the extracellular matrix. The results of X-ray microtomography and histological evaluation demonstrated that the PLGA/icariin scaffold maintained the functional morphology of articular cartilage and inhibited the resorption of subchondral bone trabeculae. These findings indicated that the PLGA and icariin composite scaffold has therapeutic potential for use in the treatment of osteoarthritis.

# 1. Introduction

Osteoarthritis (OA) is the most common musculoskeletal disease, and it is generally characterized by cartilage degradation, subchondral

bone stiffness and osteophyte formation [1]. The breakdown of cartilage, subchondral bone and other tissues in the joints causes pain and the loss of range of motion [2]. The treatments for OA are based on three critical targets: relieve pain, improve joint function and delay the progression of the disease [3].

Traditional Chinese medicine practitioners believe that there is a strong link between bones and the kidneys, and that herbs such as epimedium (*Epimedium Linn.*) improve bone health by nourishing the kidneys. Epimedium (EP) is widely used to treat arthritis in China, Japan and Korea [4]. Icariin is the main pharmacologically active molecule present in EP. Icariin stimulates osteogenic differentiation and bone formation through the MAPK signalling pathway, PI3 K-AKT signalling pathway and β-catenin signalling pathway [5–8]. A previous study demonstrated that icariin could inhibit the synthesis of nitric oxide and matrix metalloproteinase (MMPs) and could reduce the destruction of the extracellular matrix [9]. Li *et al.* confirmed that icariin could increase synthesis of the extracellular matrix and promote the restoration of supercritical-sized osteochondral defects [10]. Ma *et al.* found that icariin could attenuate hypoxia-induced apoptosis in osteoblasts [11]. However, there are few studies on the effects of icariin on osteoarthritis. Based on the effect of icariin on bone formation and cartilage matrix formation, we hypothesize that icariin can maintain the functional morphology of articular cartilage and the function of subchondral bone, which in turn prevents the progression of OA.

Intragastric administration is the traditional method for icariin administration. Local administration can deliver the drug to the site of action, thus minimizing the systemic toxic effect of the drug. To achieve the aim of prolonged and controlled release, PLGA/icariin composite scaffolds have been employed as a controlled release system. The electrospinning manufacturing of nanofibre scaffolds has a long history in the field of biomaterials, providing security, high quality and productivity [12]. According to Levorson *et al.* [13], the coculture of chondrocytes and mesenchymal stem cells on electrospun fibrous scaffolds could produce extracellular matrix for cartilage regeneration. Previous research has implied that icariin-loaded onto biomaterials might have potential for cartilage tissue engineering [14].

Therefore, in this study, icariin was incorporated into poly(lactic-co-glycolic acid) (PLGA), and PLGA/icariin composite scaffolds were prepared. The feasibility of PLGA/icariin scaffolds for tissue engineering was tested by investigating cell biocompatibility *in vitro* and the therapeutic effects on osteoarthritis *in vivo*.

# 2. Material and methods

## 2.1. Materials

The chemicals were provided by Sigma-Aldrich. PLGA (MW = 1 500 000, PLA : PGA = 75 : 25) was purchased from Changchun Sino Biomaterials Co., Ltd, and icariin powder was purchased from Chengdu Herbpurify Co., Ltd.

## 2.2. Fabrication of PLGA/icariin composite scaffold

PLGA/icariin composite scaffolds were prepared by electrospinning. The PLGA resins were dissolved in 1,1,1,3,3,3-hexafluoroisopropanol to make 10% PLGA solution supplemented with different concentrations of icariin. The concentration (w/w) of icariin was 0.01%, 0.1% or 1%. Then the mixture solution was loaded into the electrospinning apparatus with an applied voltage of 10 kV. The distance from the needle tip to the collector plate was 15 cm. The flow rate was $0.3 \, \text{ml h}^{-1}$. Subsequently, the composite scaffolds were obtained after being vacuum-dried overnight at room temperature. Finally, the fabricated spinning was sterilized with ultraviolet light for further use.

## 2.3. Physico-chemical and mechanical characterizations of PLGA/icariin composite scaffolds

The surface morphology of the PLGA/icariin composite scaffolds was observed by scanning electron microscopy (ESEM, XL30 FEG, Philips). The fibre diameter was quantified using Image-Pro Plus (version 6.0.0.260, Media Cybernetics, MD). The surface wettability of scaffolds was evaluated by the static water contact angle using a goniometer. In short, it was measured by calculating the included angle between the water droplet and horizontal plane of the scaffolds. FTIR spectroscopy was employed to detect the components of the materials in the range of $4000–600 \, \text{cm}^{-1}$, as previously described [15]. The release profile of icariin on PLGA/icariin composite fibrous scaffolds was tested by using a Tecan Infinite F 200 microplate reader at 280 nm (Crailsheim, Germany).

## 2.4. Biocompatibility of PLGA/icariin composite scaffolds

Cytocompatibility, cell morphology, an alkaline phosphate (ALP) activity assay and a real-time polymerase chain reaction (RT-PCR) assay were used to evaluate the cell responses to scaffolds *in vitro*.

### 2.4.1. Cell culture

Chondrocytes were isolated from the articular cartilage of New Zealand white rabbits (Institute of Biological Products Co., Ltd, Changchun, China). Mouse preosteoblasts (MC3T3-E1) were purchased from the Shanghai Institute of Biochemistry and Cell Biology. The cells were cultured in Dulbecco's Modified Eagle Medium supplemented with 10% fetal bovine serum (v/v), 100 U ml$^{-1}$ penicillin, and 100 mg ml$^{-1}$ streptomycin at 37°C in 24-well plates. The medium was changed every 2 days. When they reached 80% confluence, the cells were harvested for the following assessment.

### 2.4.2. Cytocompatibility

The cytocompatibility of scaffolds on chondrocytes was evaluated with an MTT method. According to different interventions, the cells were divided into eight groups: the blank control group, PLGA group, 0.01% icariin group, 0.1% icariin group and 1% icariin group, 0.01% icariin +PLGA group (w/v%, loaded in PLGA electrospinning), 0.1% icariin +PLGA group and 1% icariin +PLGA group. The cells were cultured in Dulbecco's Modified Eagle Medium supplemented with 10% fetal bovine serum (v/v), 100 U ml$^{-1}$ penicillin and 100 mg ml$^{-1}$ streptomycin at 37°C in 24-well plates. In the blank control group, phosphate-buffered saline (PBS) was added to 24-well plates. In the PLGA group and icariin +PLGA group, 1 cm$^2$ of PLGA electrospinning containing different concentrations of icariin was added to each well of the plate. In the icariin group, the same quantity of icariin was added to each well of the plate. Due to the solubility of icariin in water, the concentration of icariin in the 1% icariin group was 0.3% instead of 0.1%. Three days after, 100 μl of MTT solution (5 mg ml$^{-1}$ in PBS) was added to each well and incubated for 4 h at room temperature. The absorbance was measured using a Tecan Infinite F 200 microplate reader at 492 nm.

### 2.4.3. Cell morphology

The morphology of chondrocytes cultured with various interventions (PLGA, PLGA loading different concentrations of icariin) was observed after 24 h. The medium was removed, and the cells were washed three times with PBS. Cells were fixed with 4.0% paraformaldehyde for 12 min followed by rinsing three times with PBS. Then the cells were stained with 10% fluorescein isothiocyanate for 10 min at 37°C. Images of adhered cells were observed with a fluorescence microscope (TE2000-U, Nikon, Japan).

### 2.4.4. ALP assay

MC3T3-E1 cells were seeded at a concentration of $2 \times 10^4$ cells/well in a sterile 12-well plate in the above condition. After incubation for 3, 7 and 14 days with various interventions, the medium was removed carefully followed by rinsing three times. Two hundred microlitres of RIPA cell lysate containing 1 mM phenylmethanesulfonyl fluoride was added to each well and incubated at −80°C for 30 min. The frozen mixture was thawed at room temperature. The lysate was centrifuged for 10 min at 4°C and 10 000$g$. PNPP and BCA were added to the supernatant and maintained at 37°C for 30 min. The reaction was stopped by adding 500 μl of 1 N NaOH. The absorbance was tested on a multi-function microplate scanner at 405 and 562 nm. The ALP activity was the ratio of two measurements.

### 2.4.5. PCR of MC3T3-E1 cells

With various interventions, MC3T3-E1 cells were incubated on 24-well plates at a density of $2 \times 104$ cells per well for 7 days. Chondrogenic relative gene expressions for collagen type I (Col-I), collagen type II (Col-II), aggrecan (AGG) and osteopontin (OPN) were analysed with RT-PCR. The relative gene expressions were normalized by the expression of glycer-aldehyde-3-phosphate dehydrogenase (GAPDH). The cDNA was synthesized after the total RNA was isolated from the cells using TRIzol (Takara Biotechnology, Dalian, China). Primer sequences were designed using the Primer Premier 5.0 software (table 1). RT-PCR reactions were performed using a SYBR Green Mix Kit (Takara Biotechnology). The PCR amplification cycles were carried out as follows: denaturation at 95°C for 5 s,

**Table 1.** Primer nucleotide sequences.

| gene | primer nucleotide sequences (5′−3′) |
| --- | --- |
| AGG | F: CAAGGACAAGGAGGTGGTG |
| | R: GTAGTTGGGCAGCGAGAC |
| OPN | F: CGTGGATGATATTGATGAGGATG |
| | R: TCGTCGGAGTGGTGAGAG |
| Col-I | F: CTCGCTCACCACCTTCTC |
| | R: TAACAACTGCTCCACTCTG |
| Col-II | F: CTCAAGTCCCTCAACAACC |
| | R: AGTAGTCACCGCTCTTCC |
| GAPDH | F: GATGGTGAAGGTCGGAGTG |
| | R: TGTAGTGGAGGTCAATGAATGG |

annealing, and extension for 30 s at 56°C for 40 cycles. The reaction was performed using a StepOnePlus Real-Time PCR System (Applied Biosystems, CA, USA).

## 2.5. Treatment of rabbit OA

All animal experiments were carried out in accordance with the United States National Institutes of Health Guide for the Care and Use of Laboratory Animals. The experimental procedures used for this study were reviewed and approved by the affiliated hospital to Changchun University of Chinese Medicine (CCZYFYLL 2017 SHEN- 057). Twenty New Zealand white rabbits were randomly divided into two groups: the PLGA group and the PLGA/icariin group. Models of osteoarthritis were induced as previously described [3]. The New Zealand white rabbits were anaesthetized by injection of 10% chloral hydrate (4 ml kg$^{-1}$). Following anaesthetic induction, models of osteoarthritis were performed in New Zealand white rabbits as follows. Right knee joints were explored through an anterior medial incision. The surgical progress included medial collateral ligament transection, anterior cruciate ligament transection, and meniscectomy. The capsule and skin were sutured with 5-0 absorbable surgical sutures (Suzhou Medical Appliance Factory, China). After surgery, penicillin potassium (3 million U/kg, Sichuan Huanya Biotechnology, China) was injected for three days. The rabbits were housed in individual cages at 21 ± 0.5°C. The rabbits had free access to food and water. Four weeks later, a rabbit knee osteoarthritis model was successfully induced. The New Zealand white rabbits were anaesthetized, and the PLGA, PLGA/icariin scaffolds were implanted into the right knees of rabbits. According to the results *in vitro*, the concentration of icariin *in vivo* was 0.1%. The scaffolds were folded and filled with the defect area caused by the meniscus being removed. Then the rabbits were treated as in the first post-operation period.

## 2.6. X-ray microtomography examination

The microarchitecture of trabecular bone in distal femurs was analysed at the fourth week, eighth week and 12th week after surgery using X-ray microtomography (XMT) scanning (SkyScan 1172, SkyScan, Belgium). After the rabbits were sacrificed, the right knees were isolated and fixed using 10% formaldehyde. The function parameter of the micro CT was set as 80 kV and 100 µA, and the thickness of serial tomographic images was set as 18 µm. The outcomes of bone structure were calculated by CTAn (CTAn, SkyScan, Belgium) according to the manufacturer's instructions.

## 2.7. Histological observation and evaluation

For histological analysis of articular cartilage and subchondral bone, the fixed right knee was decalcified in 10% EDTA for approximately eight weeks. Then the dehydrated bones were cut into 4 µm thick sections for histomorphometry. After haematoxylin and eosin (HE) and toluidine blue (TB) staining, the sections were observed with an Olympus CX31 (Olympus, Japan) microscope. Histological

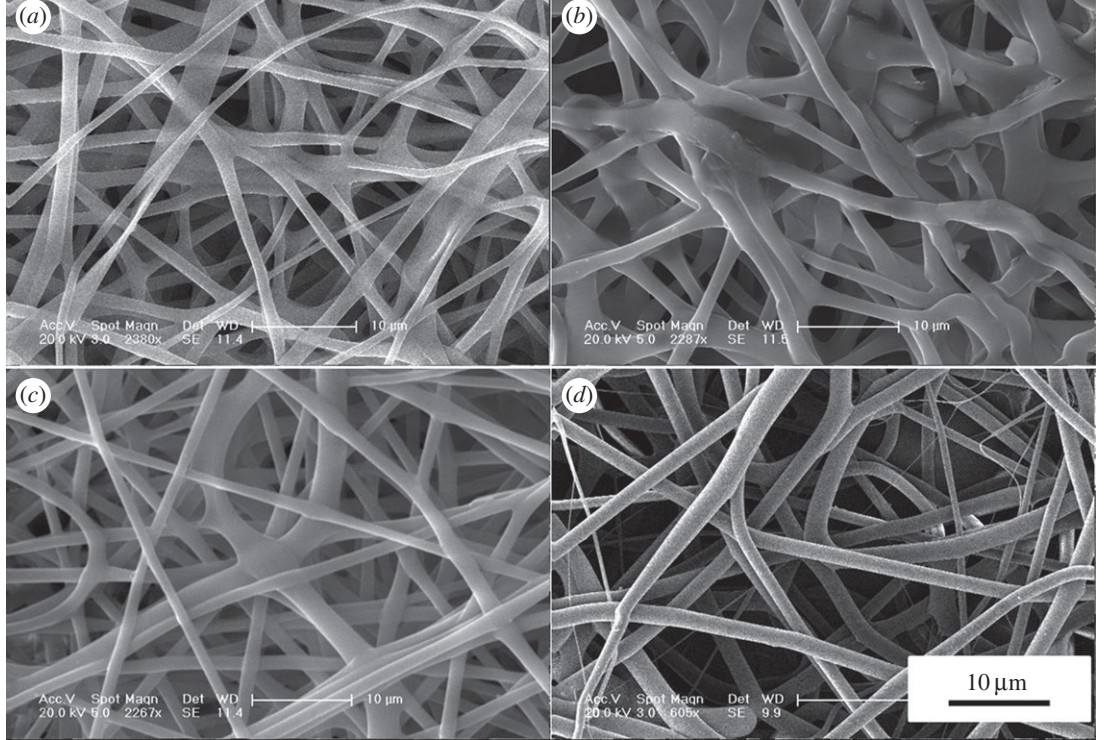

**Figure 1.** SEM images of the fibrous scaffolds. (*a*) PLGA fibrous scaffold. (*b*) PLGA/0.01% icariin fibrous scaffold. (*c*) PLGA/0.1% icariin fibrous scaffold. (*d*) PLGA/1% icariin fibrous scaffold. Bar lengths are 10 μm.

grading was performed according to a modified Mankin scoring system established for grading OA changes, as in a previous study [1].

## 2.8. Real-time polymerase chain reaction assay of articular cartilage

To further study the effect of icariin on the synthesis of extracellular matrix in articular cartilage, a RT-PCR assay was performed at the fourth week and eighth week after intervention. After the frozen tissue was crushed, the total RNA was isolated from the articular cartilage using TRIzol. The following procedure was the same as in the previous RT-PCR assay of mesenchymal stem cells.

## 2.9. Statistical analysis

IBM SPSS 21.0 software (SPSS Inc., Chicago, IL, USA) was used for statistical comparisons. The evaluation of effectiveness was compared with one-way analysis of variance (ANOVA). The same analysis parameters were used for *t*-tests. *p*-values < 0.05 were considered statistically significant.

# 3. Results

## 3.1. The characterization of fibrous scaffolds

The surface morphology and microstructure of the fibrous scaffolds were investigated using SEM (figure 1). The mean diameter of the PLGA fibrous scaffolds was $1.39 \pm 0.31$ μm. After adding different concentrations of icariin, the surface morphology of the fibres became rough. The mean diameters of composite fibrous scaffolds were $1.99 \pm 0.74$, $1.42 \pm 0.50$ and $1.55 \pm 0.63$ μm. The mean diameter of fibrous scaffolds became thicker and the thickness was uneven with low concentrations of icariin. The results showed that the presence of icariin influenced the fibre diameter at low concentrations. As the concentration of icariin increased, the diameter of the spinning tended to decrease and reduce in roughness. The contact angle results indicated the hydrophilicity of the materials. The hydrophilicity of the materials plays an important role in interacting with cells. The

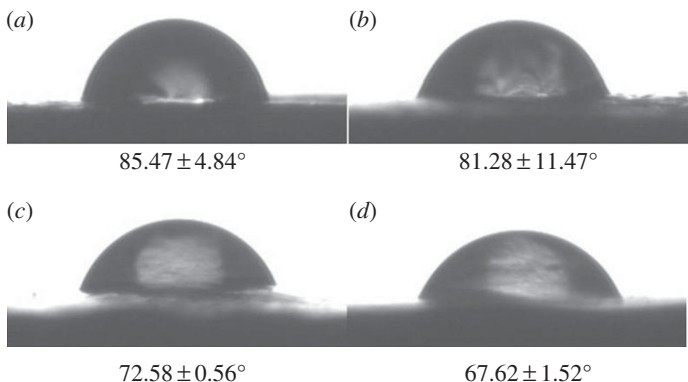

**Figure 2.** The contact angles of the fibrous scaffolds. (*a*) PLGA fibrous scaffold. (*b*) PLGA/0.01% icariin fibrous scaffold. (*c*) PLGA/0.1% icariin fibrous scaffold. (*d*) PLGA/1% icariin fibrous scaffold. The contact angle of pristine PLGA was $85.47 \pm 4.84°$, and it decreased gradually from $81.28 \pm 11.47°$ to $67.62 \pm 1.52°$ with the gradual increase in icariin concentration.

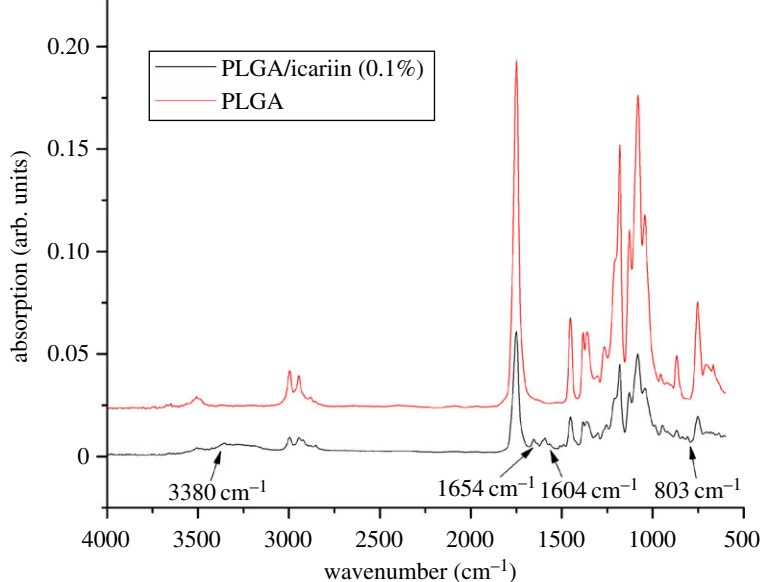

**Figure 3.** The infrared spectra of composite scaffolds. The red line is PLGA, and the black line is PLGA/icariin. The arrows show four characteristic bands at 803, 1604, 1654 and 3380 $cm^{-1}$ in PLGA/icariin.

contact angle of pristine PLGA was $85.47 \pm 4.84°$, and it decreased gradually from $81.28 \pm 11.47°$ to $67.62 \pm 1.52°$ with gradual increases of icariin concentrations (figure 2). The FTIR spectroscopy was in ATR mode and the infrared spectra of composite scaffolds are shown in figure 3. Compared to pristine PLGA, there were four characteristic brands in PLGA/icariin: 803, 1604, 1654 and 3380 $cm^{-1}$. In the release experiment of icariin, PLGA/icariin was soaked in PBS, and the optical density of the supernatant was measured at 280 nm. The investigation of the icariin release profile showed that there was a burst release within 12 h in the three groups (figure 4). The release subsequently slowed down, and there was still icariin release until 192 h. Among the three concentrations, the scaffolds with 0.1% icariin had a slower release rate.

## 3.2. Cytocompatibility and cell morphology

Compared to the blank control (figure 5), the proliferation of cells cultured with icariin varied significantly with different concentrations of icariin ranging from 0.01% to 1% ($p < 0.05$). Compared to the blank control, icariin could promote the proliferation of chondrocytes ($p < 0.05$). With an increase in the concentration, the promotion ability rose and then fell. Compared to pristine PLGA,

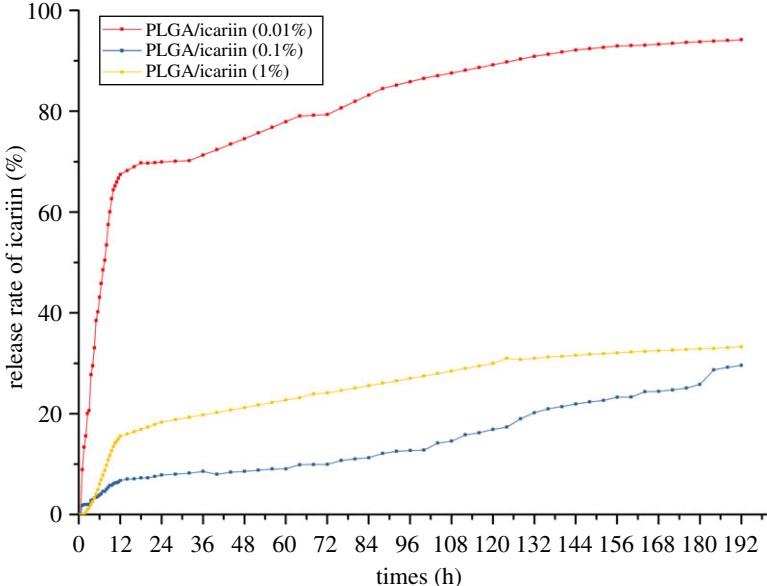

**Figure 4.** Cumulative release of icariin from PLGA/icariin composite scaffolds at different time points. The release profile of icariin showed that there were burst releases from the PLGA/icariin composite scaffolds within 12 h in three groups, and the release subsequently slowed down and maintained icariin release until 192 h. In the PLGA/0.01% icariin group, 67.5% of the loaded drug had been released after 12 h, while 7.2% of the loaded drug had been released in the PLGA/0.1% icariin group. The PLGA/0.1% icariin group showed a sustained and slow release.

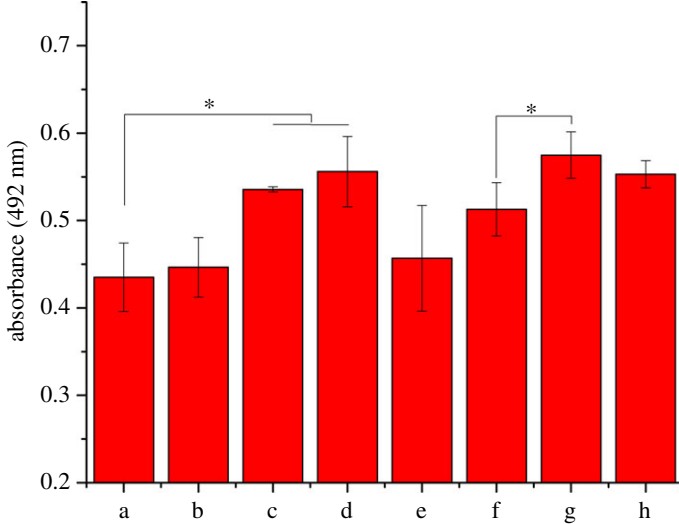

**Figure 5.** Chondrocyte cytocompatibility assay. (a) Control group. (b) PLGA fibrous scaffold. (c) 0.01% icariin group. (d) 0.1% icariin group. (e) 1% icariin group (the actual drug concentration is 0.3%). (f) 0.01% icariin + PLGA fibrous scaffold. (g) 0.1% icariin + PLGA fibrous scaffold. (h) 1% icariin + PLGA fibrous scaffold. Compared to the blank control, icariin could promote the cytocompatibility of chondrocytes ($p < 0.05$). With the increase in the concentration, the promotion ability rose and then fell. Compared to pristine PLGA, PLGA/icariin scaffolds could promote the proliferation of chondrocytes ($p > 0.05$). There were no significant differences among the three icariin concentrations ($p > 0.05$), *$p < 0.05$.

PLGA/icariin scaffolds could promote the proliferation of chondrocytes ($p > 0.05$). There were no significant differences among the three icariin concentrations ($p > 0.05$). Chondrocytes were fluorescently stained to investigate cell adhesion and migration behaviours. After 24 h of culture, fluorescence microscopy observations demonstrated a significant increase in the number of adhered chondrocytes cells in the presence of icariin compared to untreated controls (figure 6). The cell spreading areas cultured with PLGA/icariin were wider than those cultured with PLGA.

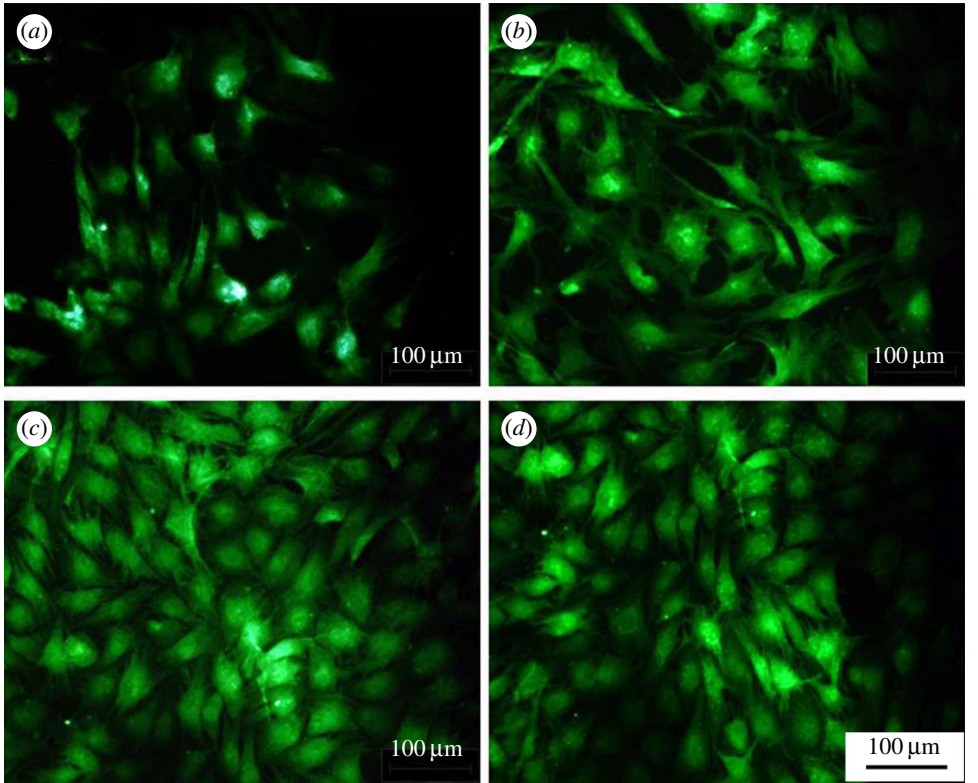

**Figure 6.** Chondrocyte attachment assay. (*a*) PLGA fibrous scaffold; (*b*) PLGA/0.01% icariin fibrous scaffold; (*c*) PLGA/0.1% icariin fibrous scaffold; (*d*) PLGA/1% icariin fibrous scaffold. The number and spreading areas of adhered chondrocytes cells cultured with PLGA/icariin were better than those cultured with PLGA. Bar lengths are 100 μm.

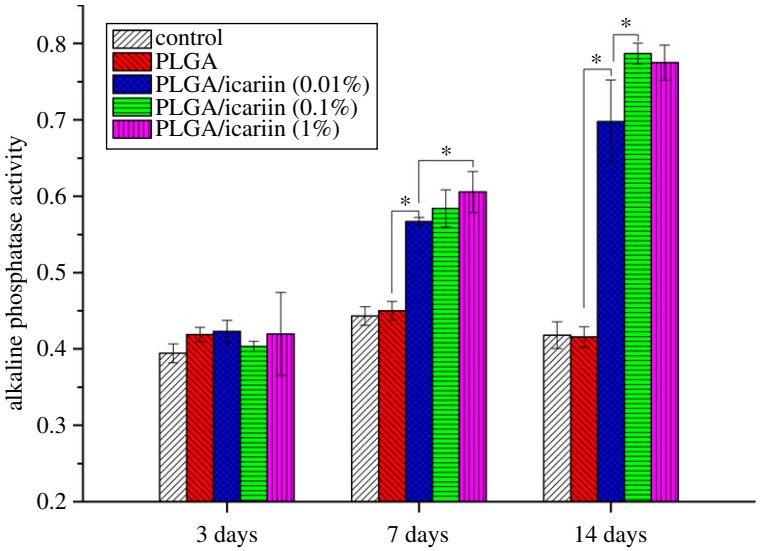

**Figure 7.** ALP activity. ALP activity was quantified in culture supernatants at 3 days, 7 days and 14 days. The ALP activity of MC3T3-E1 cells cultured with PLGA/icariin scaffolds was significantly higher than that cultured without icariin 7 days later ($p < 0.05$). After 14 days of culture, the ALP activity was better than that cultured for 7 days, *$p < 0.05$.

## 3.3. ALP activity assay

There was no significant difference 3 days later in the ALP activity of MC3T3-E1 cells cultured with different scaffolds (figure 7). However, the ALP activity of MC3T3-E1 cells cultured with PLGA/icariin scaffolds was significantly higher 7 days later than that of cells cultured without icariin ($p < 0.05$). There was a statistical difference in ALP activity between 1% icariin and 0.1% icariin ($p < 0.05$). After 14 days of culture, the ALP

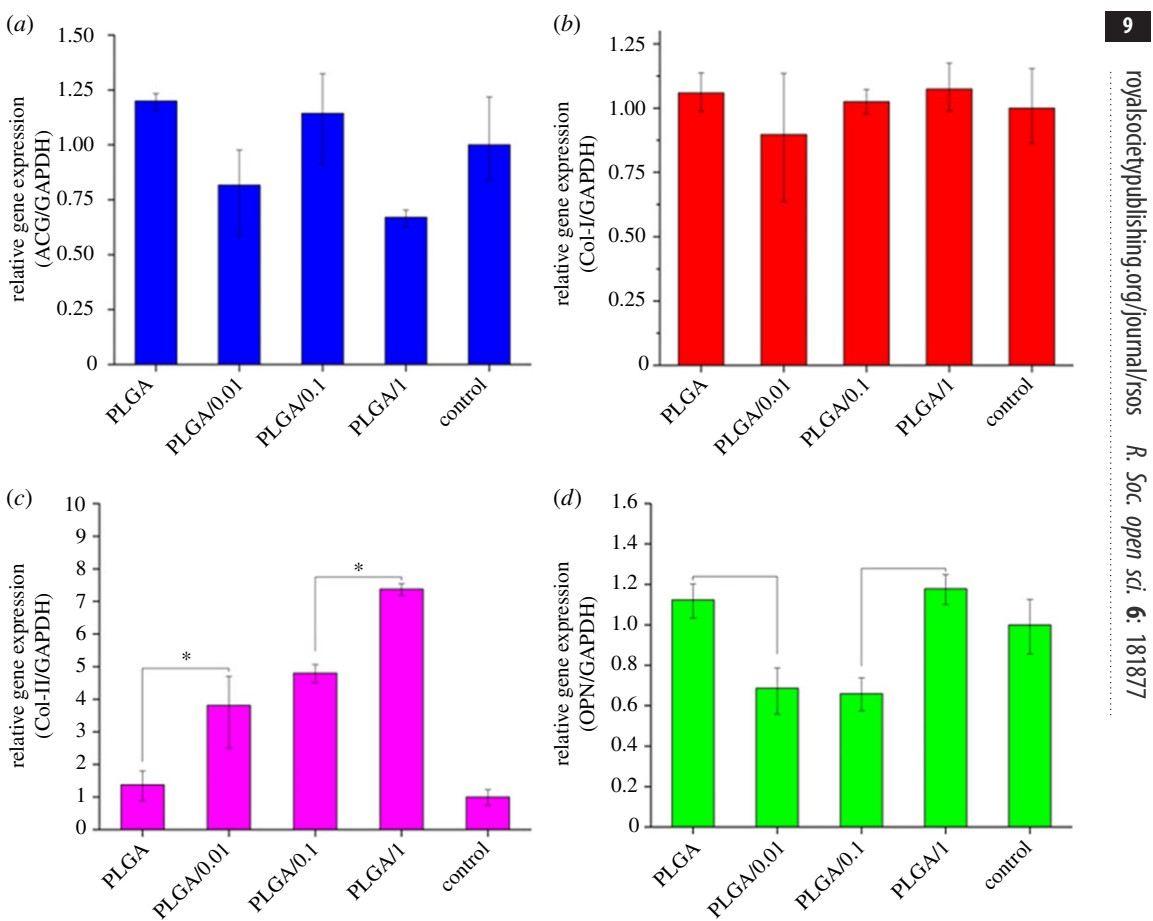

**Figure 8.** Real-time polymerase chain reaction assay of MC3T3-E1 cells. (*a*) The expression of AGG. (*b*) The expression of Col-I. (*c*) The expression of Col-II. (*d*) The expression of OPN. *$p < 0.05$.

activity of MC3T3-E1 cells cultured with PLGA/icariin was better than that of cells cultured for 7 days, but there was no statistical difference at three time points in MC3T3-E1 cells cultured without icariin, indicating that icariin could induce MC3T3-E1 cell osteogenic differentiation.

## 3.4. Real-time polymerase chain reaction assay of MC3T3-E1 cells

The expression of AGG was not promoted by PLGA or PLGA/icariin scaffolds (figure 8*a*). Moreover, at the concentration of 1%, icariin led to downregulation of the expression of AGG. There were no significant differences in Col-I among the groups (figure 8*b*, $p > 0.05$). The results indicated that PLGA or PLGA/icariin scaffolds could not induce the expression of AGG and Col-I in mesenchymal stem cells. The two scaffolds promoted the expression of Col-II in mesenchymal stem cells and the promotion increased with increasing icariin concentrations (figure 8*c*, $p < 0.05$). At the concentration of 0.01% and 0.1%, icariin led to downregulation of the expression of OPN (figure 8*d*, $p < 0.05$). However, PLGA and PLGA/1% icariin did not have a significant effect on the expression of OPN.

## 3.5. Histological observation and evaluation

The morphology of knees in the PLGA/icariin group was better that in the PLGA group (figure 9*a*). The modified Mankin scores of the PLGA group and PLGA/icariin group four weeks after surgery were $5.97 \pm 1.82$ and $5.67 \pm 1.28$, respectively (figure 9*b*). There was no significant difference between the two groups ($p > 0.05$). The modified Mankin score in the PLGA group was significantly higher than in the PLGA/icariin group at eight weeks and 12 weeks post-surgery ($p < 0.05$). The chondrocytes were arranged in a straight line in the PLGA/icariin group, while the chondrocytes in the PLGA group were clustered (figure 9*c*). The proteoglycan stained with toluidine blue was gradually reduced in both groups. The thickness of the subchondral bone plate was thicker in the PLGA group, which might mean that the channels were narrower (figure 9*c*).

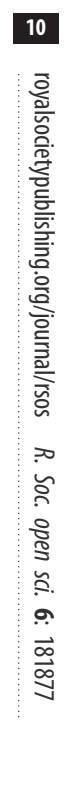

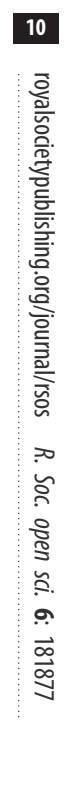

**Figure 9.** Histological observation and evaluation. (*a*) Distal femur morphology (a: the morphology of distal femur pre-implantation; b: the morphology of distal femur four weeks after implantation; c: the morphology of distal femur eight weeks after implantation; d: the morphology of distal femur 12 weeks after implantation). (*b*) The modified Mankin scores; the modified Mankin score in the PLGA group was significantly higher than that of the PLGA/icariin group at eight weeks and 12 weeks post-surgery ($p < 0.05$), *$p < 0.05$. (*c*) The chondrocytes were arranged in a straight line in the PLGA/icariin group, while the chondrocytes in the PLGA group were clustered, and the proteoglycan stained with toluidine blue was gradually reduced in both groups (e: HE stain of PLGA group; f: HE stain of PLGA/icariin group; g: toluidine blue stain of PLGA group; h: toluidine blue stain of PLGA/icariin group). Bar lengths are 200 μm.

## 3.6. Real-time polymerase chain reaction assay of articular cartilage

The expression of AGG was promoted by PLGA/icariin scaffolds at four weeks after the intervention, while it was inhibited by PLGA scaffolds (figure 10*a*, $p < 0.05$). The results indicated that icariin can promote the expression of AGG. There were no significant differences between PLGA and PLGA/icariin scaffolds in the synthesis of Col-I (figure 10*b*, $p > 0.05$). PLGA/icariin scaffolds could promote the expression of Col-II, while PLGA scaffolds could not promote this expression (figure 10*c*, $p < 0.05$). The promotion lasted until eight weeks after the intervention. The results indicated that icariin can promote synthesis of

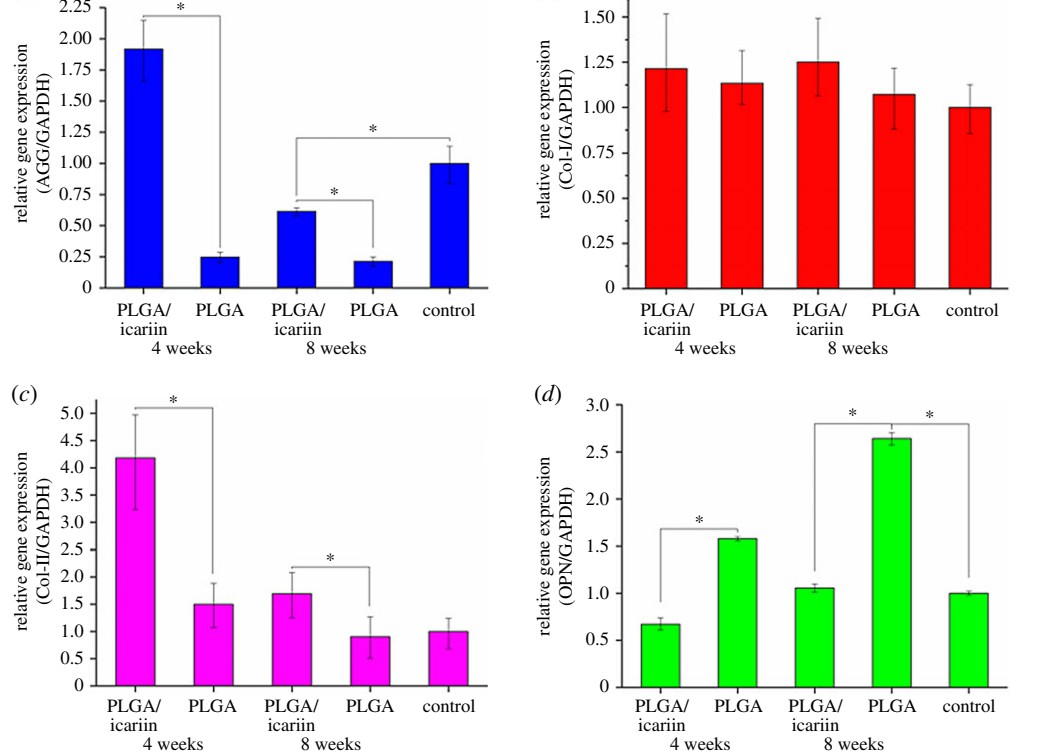

**Figure 10.** PCR of articular cartilage. (a) The expression of AGG was promoted by PLGA/icariin scaffolds at four weeks after intervention, while AGG expression was inhibited by PLGA scaffolds ($p < 0.05$). (b) There were no significant differences between PLGA and PLGA/icariin scaffolds in the synthesis of Col-I ($p > 0.05$). (c) PLGA/icariin scaffolds could promote the expression of Col-II. (d) PLGA/icariin inhibited OPN expression while PLGA promoted its expression ($p < 0.05$), *$p < 0.05$.

the extracellular matrix and maintain the functional integrity of cartilage. PLGA/icariin inhibited OPN expression while PLGA promoted its expression (figure 10d, $p < 0.05$). From the fourth week to the eighth week, the expression of OPN gradually increased in both groups ($p < 0.05$).

## 3.7. XMT examination

Both the PLGA group and the PLGA/icariin group exhibited a decrease in the trabecular thickness (Tb.Th) and the trabecular number (Tb.N) after intervention. Compared to the PLGA group, the change in Tb.Th and Tb.N of the PLGA/icariin group was significant (figure 11). The area in the red line is the region of interest for the XMT test. There were more osteophytosis around the joints. These results illustrated icariin can promote the formation of trabecular bone.

# 4. Discussion

The findings of this study demonstrated that icariin incorporated into electrospun fibres could improve the hydrophilicity of the scaffold, maintain a slow and steady release and have good biocompatibility *in vitro*. Compared with a single PLGA scaffold, the PLGA/icariin scaffold could maintain the functional morphology of articular cartilage and inhibit the resorption of subchondral bone trabeculae. Together, these data suggest the PLGA and icariin composite scaffold has therapeutic potential for use in the treatment of OA.

In this study, we loaded icariin into PLGA electrospinning. The SEM showed that there was a statistical change in the average fibre diameter at low concentrations. With the increase in drug concentrations, the diameter gradually became thinner and more uniform. The diameter of nanofibres can be affected by fabrication parameters, such as the orifice diameter, solution concentration and voltage per unit length. When the concentration of the solution is low, the effect of surface tension is greater than that of the viscoelastic force, and the diameter of the fibre is unstable or relatively rough [16,17]. The results indicated that the addition of 0.1% icariin had little effect on the concentration of

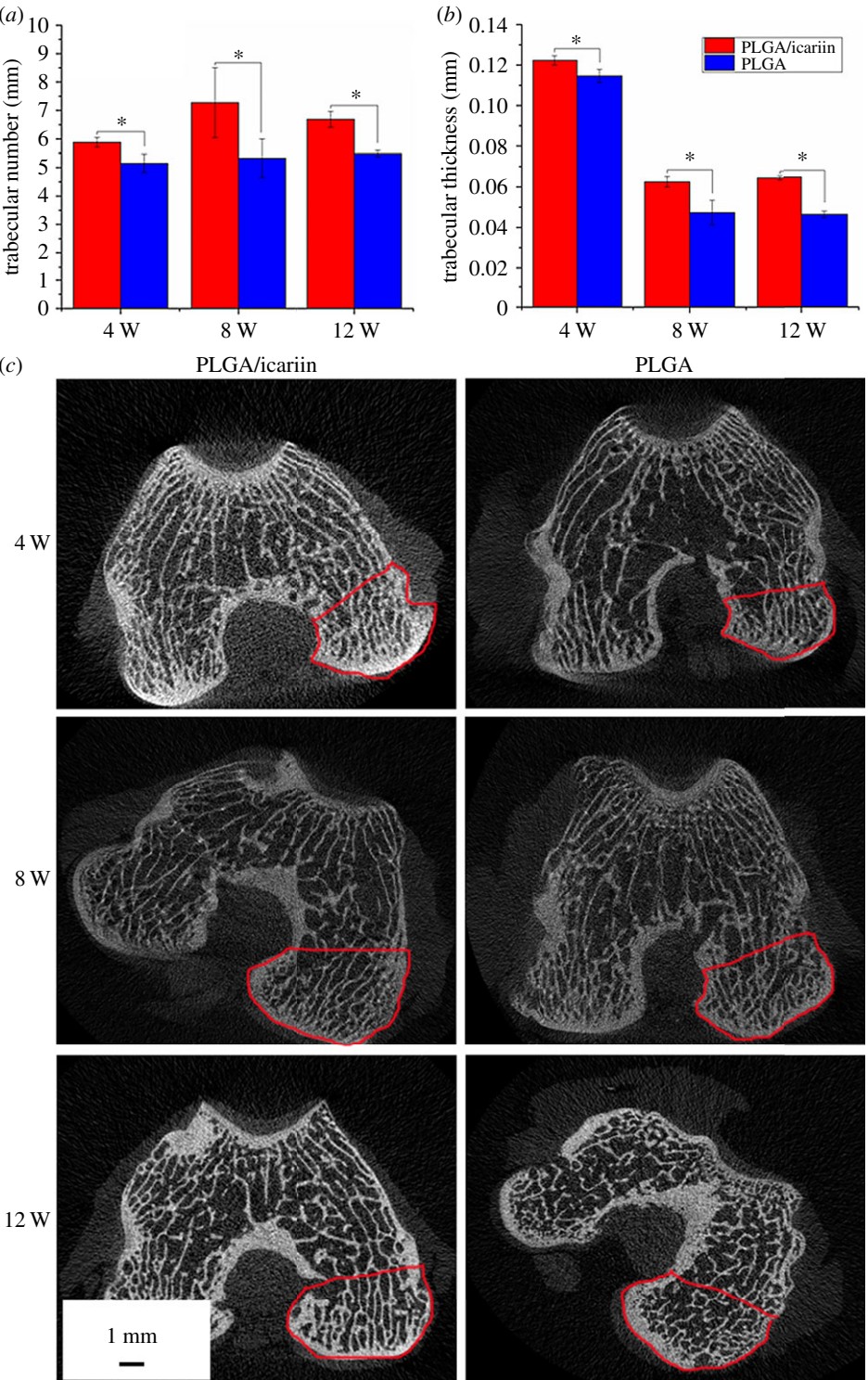

**Figure 11.** X-ray microtomography examination. (*a*) Trabecular number. (*b*) Trabecular thickness. *$p < 0.05$. Both the PLGA group and the PLGA/icariin group exhibited a decrease in Tb.Th and Tb.N, and the decline in levels of the PLGA/icariin group was slower. (*c*) CT scans of the distal femur; there were more hyperplastic bones around the joints in the PLGA group. The area in the red line is the region of interest for the X-ray microtomography test.

the solution. The results of the contact angle indicated that the incorporation of icariin could improve the hydrophilicity of the fibrous scaffolds. Since the hydrophilicity of the scaffold plays a key role in interacting with cells, it promotes cell adhesion and proliferation. The above results showed that the fibrous scaffolds can provide a suitable microenvironment for cells and promote the attachment and proliferation of cells. The characteristic absorption bands of EP were at 3380, 2918, 2850, 1734, 1654,

1604, 1385, 1257, 1054 and 1033 cm$^{-1}$ [15]. The region between 1800 and 1500 cm$^{-1}$ of the spectra corresponds to infrared absorption by the carboxylic ester, and the region between 800 and 1200 cm$^{-1}$ of the spectra corresponds to carbohydrate [18]. Vibrations of O-H are typically found around the 3432 cm$^{-1}$ region [19]. The wavenumbers at 1604 cm$^{-1}$ correspond to the stretching vibration of total flavonoid bonds [15]. The results showed that the characteristic absorption bands were 803, 1604, 1654 and 3380 cm$^{-1}$, which indicated that there was icariin in the PLGA/icariin composite scaffolds. The fast release of icariin leads to high dose requirements and short durations of validity and even more risk of side effects. The release of the drug from scaffolds might be related to initial drug loading, drug solubility, drug diffusion, drug/scaffold interactions and scaffold degradation [20]. In this study, icariin was encapsulated and stored in the scaffold. During the first 12 h, the icariin was superficially released, which led to a burst release. As time went on, icariin that was encapsulated in the scaffold could only be released slowly. The release experiment result demonstrated that PLGA/icariin composite fibrous scaffolds could effectively immobilize icariin, and it was released slowly and steadily from the scaffold. The result is consistent with a previous study [21].

The present study showed the PLGA/icariin scaffolds had good biocompatibility. It was found that the suitable concentration for stimulating the proliferation and osteogenic differentiation of human bone marrow stromal cells ranged from $10^{-9}$ to $10^{-6}$ M [22]. In the present study, the cells cultured with icariin solution showed chondrocyte proliferation in an icariin-dose dependent manner. At icariin concentrations of approximately 0.3%, the absorbance decreased, which may due to the cytotoxicity of icariin. However, when it was cultured with a PLGA/icariin scaffold, they all could promote the proliferation of chondrocytes, and there was no significant difference. The possible reason is that the slow and steady release of icariin leads to a stable concentration of icariin in the medium. The results of the cell adhesion assay were consistent with the MTT assay. The results supported PLGA/icariin composite scaffolds having good biocompatibility. Electrospun fibrous scaffolds can accelerate proliferation of chondrocytes and maintain the chondrocytic phenotype in vitro [23,24]. The PLGA/icariin composite scaffolds have better promoting effects than PLGA fibrous scaffolds.

A previous study showed that ALP activity was low at day 4 and at day 10 (the maturation and mineralization periods) in MC3T3-E1 cells [25]. In our study, there was not a dose-dependent effect after the cells were cultured for 3 days. The ALP activity was significantly increased at the concentrations of 0.1% and 1% after the cells were cultured for 7 days and 14 days. The results of the two studies are consistent. ALP is a marker of the pre-osteoblast stage of differentiation [26]. The results of ALP activity suggested that icariin can promote the pre-osteoblast difference in MC3T3-E1. The mRNA expression of Col-II and AGG was defined as a marker of chondrocyte differentiation [27]. Although PLGA/icariin can promote the expression of Col-II, it cannot promote the expression of AGG. The results of PCR do not support the PLGA/icariin scaffold inducing chondrocytic differentiation of MC3T3-E1.

Cartilage destruction is one characteristic of OA. The evaluation of OA can be performed through the analysis of five characteristics: the structure of cartilage, chondrocyte number, chondrocyte clustering, proteoglycan content, and integrity of the tidemark. A previous study showed there were channels on the subchondral bone plate, which could provide a direct link between articular cartilage and subchondral trabecular bone, and the shape and diameter of the channels could be affected by the thickness of the subchondral bone plate [28]. The modified Mankin score and the thickness of the subchondral bone plate suggested that icariin can delay the degeneration of cartilage, thereby delaying the pathological changes of OA.

Proteoglycans and collagens are two major components in ECM that can maintain the functional integrity of cartilage [29]. The synthesis of the two components suggests that icariin can promote the synthesis of ECM in chondrocytes. The function of articular cartilage relies on the structural integrity and biochemical composition of the ECM, i.e. mainly collagen and proteoglycan. The present study proved that PLGA/icariin could promote the synthesis of Col-II and AGG in vivo and then maintain the structural integrity of cartilage. OPN was synthesized by bone-forming cells and hypertrophic chondrocytes, and a previous study suggested that OPN mRNA was found in cartilage from patients with osteoarthritis, while no OPN mRNA expression was observed in chondrocytes of adult healthy cartilage [30,31]. This result means that icariin inhibits terminal differentiation of chondrocytes to hypertrophic cells.

Articular cartilage and subchondral bone act as a functional unit during joint movements [32]. Subchondral bone is the key factor causing cartilage deterioration in OA, and the features of subchondral can be considered as biomarkers of OA [33]. A previous study attenuated the TGF-β signalling of articular cartilage and subchondral bone, and the results showed that the homeostasis microenvironment could be potential therapies for OA [32]. Another study showed that icariin could inhibit TGF-β canonical Smad signalling in glomerular mesangial cells [34]. It is possible that the

TGF-β pathway is the mechanism of icariin in regulating the microenvironment of cartilage and subchondral bone.

OA is a group of diseases that affects all tissues within the joint in middle-aged and elderly patients and is the result of mechanical and biological factors [35]. Reduced bone mineral density, stiffness and hardness of the subchondral bone have been observed in different stages of OA [36,37]. A previous study demonstrated mechanical properties of subchondral bone affected the integrity of articular cartilage [38]. Icariin could promote bone formation and prevent osteoporosis in postmenopausal women and in an animal model [4,39–41]. Experimental studies suggested the impact on bone metabolism could affect the progression of OA [42]. The data of Wei *et al.* indicated that icariin suppressed articular bone loss and prevented joint destruction in rheumatoid arthritis [43]. In this study, we confirmed that PLGA/icariin scaffolds had good biocompatibility during *in vitro* experiments. Although icariin cannot induce chondrogenic differentiation of MC3T3-E1 *in vitro*, it can promote the synthesis of ECM *in vivo*. Therefore, we predicted that there is a different mechanism of icariin *in vitro* and *in vivo*. Furthermore, we predicted that icariin can effectively inhibit the progression of OA. The hypothesis is consistent with our findings that PLGA/icariin scaffolds can promote the synthesis of ECM, maintain the functional morphology of articular cartilage, and inhibit the absorption of subchondral bone trabeculae. Taken together, these data suggest that the PLGA and icariin composite scaffold has therapeutic potential for use in the treatment of OA. However, the results of RT-PCR did not support the icariin-induced differentiation of mesenchymal stem cells into chondrocytes. In future work, we will improve the scaffold to facilitate implantation and load the cytokines that promote differentiation into chondrocytes.

# 5. Conclusion

In this study, a PLGA/icariin scaffold was prepared by electrospinning. The loading of icariin can reduce the hydrophilicity of the spinning scaffold, and icariin bound on the scaffold can be slowly released *in vitro*. *In vitro* and *in vivo* studies indicated that the PLGA/icariin scaffold possessed excellent biocompatibility, promoted the synthesis of ECM *in vivo*, maintained the functional morphology of articular cartilage, and inhibited the absorption of subchondral bone trabeculae. These findings indicated that the PLGA and icariin composite scaffold has therapeutic potential for use in the treatment of OA.

Ethics. All animal experiments were carried out in accordance with the United States National Institutes of Health Guide for the Care and Use of Laboratory Animals. The experimental procedures used for this study were reviewed and approved by the affiliated hospital of Changchun University of Chinese Medicine.

Data accessibility. This article does not contain any additional data.

Authors' contributions. S.J.L., J.A.L. and T.T.H. carried out all the experiments, analysed the data, interpreted the results and wrote the manuscript. C.F.Z., Z.H.L. and Y.W. performed the sample preparation and prepared all samples for analysis. C.F.Z. supervised all the experiments. All authors gave their final approval for publication.

Competing interests. We have no competing interests.

Funding. This work was supported by Jilin Provincial Science and Technology Plan of China (20160101027JC).

Acknowledgements. The support and cooperation of all the participants is appreciated.

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
