## [Reviewer comments · Royal Society Open Science]

Review History

RSOS-181159.R0 (Original submission)

Review form: Reviewer 1

Is the manuscript scientifically sound in its present form?

Yes

Are the interpretations and conclusions justified by the results?

Yes

Is the language acceptable?

Yes

Is it clear how to access all supporting data?

Not Applicable

Do you have any ethical concerns with this paper?

No

Have you any concerns about statistical analyses in this paper?

No

Recommendation?

Reject

Comments to the Author(s)

Comments to the authors:

Previous studies have indicated that Icariin, a component extracted from epimedium, enhances synthesis of cartilage extracellular matrix (ECM) in vivo. However, it remains largely unclear about possible effect of Icariin on pathological conditions of osteoarthritis (OA). In this paper, the authors found that Icariin was released steadily from PLGA/Icariin scaffold. Next, proliferation of chondrocytes on PLGA/Icariin scaffold, but not PLGA scaffold only, was enhanced. Further, implantation of the material into OA model rabbits resulted in increased amount of cartilage ECM. I think that this manuscript contains potentially interesting observation. However, I'm afraid that the results seem to be low impact (see below). Thus, I cannot recommend the acceptance of this manuscript in Royal Society Open Science.

Major concerns:

1. It has been already reported that Icariin promotes synthesis of cartilage ECM as well as osteogenesis. I cannot find a novelty of result of this study.
2. They found that PLGA/Icariin, but not PLGA only, enhances proliferation of chondrocytes on PLGA scaffold (Fig 5). However, Icariin suppresses chondrocyte differentiation via inhibition of TGF- β signaling (Li YC et al., Clin Exp Pharmacol Physiol 2013; 40; 635-43). Thus, it might be inappropriate to use Icariin as therapy of OA. Further, the result didn't show chondrocyte proliferation in an Icariin-dose dependent manner. Unfortunately, the result won't show role of Icariin in chondrocyte proliferation. I suggest you will try this study in lower concentration.

Minor concerns:

1. It would be desirable to study Icariin release in the different concentrations (Fig4). The study would enhance quality of data.
2. They must show pictures of affected tissues stained with HE or Toluidin blue before the material implantation (Fig 8). It would be an important point to study effect of the implantation on phenotypes by comparing between pre- and post-implantation.
3. There're many typos through the whole manuscript. 'Inhibited' should be 'inhibit' (lane 2 in page 2). 'The express' should be 'the expression' (line 43 in page 5). 'In previous study show' should be 'A previous study showed' (line 5 in page 7). As you know, a word described at first time in the manuscript should be formal name. In this case, OPN should be Osteopontin (line 18, page 7). Transcripts should be as italic (line 21 in page 7). 'diseases affects' should be 'disease that affects' (line 35 in page 7). 'the progress' should be 'the progression' (line 42 in page 7). 'the data of' should be 'the data by' (line 43 in page 7). The fonts are different (ref no 28 and 36). Pages are incomplete (ref no 7, 8, 15, 36, 40). Title should be indicated as upright type (ref no 29).

Review form: Reviewer 2**Is the manuscript scientifically sound in its present form?**

Yes

Are the interpretations and conclusions justified by the results?

Yes

Is the language acceptable?

Yes

Is it clear how to access all supporting data?

Yes

Do you have any ethical concerns with this paper?

No

Have you any concerns about statistical analyses in this paper?

No

Recommendation?

Accept as is

Comments to the Author(s)

In this manuscript, icariin was released by the poly (lactic-co-glycolic acid (PLGA) electrospun delivery system for prevention of osteoarthritis. Icariin was released slowly and steadily from the scaffold in addition to enhanced chondrocyte proliferation and adhesion. Authors showed sufficient evidence that the icariin has improved the osteogenesis in addition to the previous work shown by using similar electrospun polymer networks.

Review form: Reviewer 3

Is the manuscript scientifically sound in its present form?

Yes

Are the interpretations and conclusions justified by the results?

Yes

Is the language acceptable?

Yes

Is it clear how to access all supporting data?

Yes

Do you have any ethical concerns with this paper?

No

Have you any concerns about statistical analyses in this paper?

No

Recommendation?

Accept with minor revision (please list in comments)

Comments to the Author(s)

The manuscript from Wang et al. entitled "PLGA scaffold loaded with icariin for inhibiting the progression of osteoarthritis in rabbits" presents the effect of Icariin from PLGA scaffolds on inhibiting the progression of osteoarthritis. The article mainly demonstrated the feasibility of

PLGA/Icariin scaffolds for tissue engineering, which was tested by investigating cell biocompatibility *in vitro*, and the therapeutic effects on osteoarthritis *in vivo*. The article was concise and well written. The article could be published after addressing these comments. The introduction should be improved with sufficient comparisons relevant to other technologies such as supercritical fluid technology and 3D printing, which have been utilized for the generation of scaffolds and delivery of active substances for cell growth. I suggest citing these articles relevant to these technologies. *Adv. Healthcare Mater.* 2017, 1700433, *International Journal of Nanomedicine* 2018:13 4227–4245, *J. of Supercritical Fluids* 120 (2017) 43–51, *International Journal of Nanomedicine* 2017:12 1877–1890, *Biofabrication* 9 (2017) 032002, *ACS Biomater. Sci. Eng.* 2018, 4, 800–818, *Materials* 2018, 11, 1390, *Polymers* 2018, 10, 807, *Nanomaterials* 2018, 8, 360.

Better input the respective wavenumber in the graph (Fig. 3) rather than in the figure legend. Loading amount of Icariin?, please represent the release amount in the form of percentage in Figure 4.

I do not think chondrocyte proliferation concerning that showing the effect of Icariin is necessary, as the aim is towards inhibiting the progression of arthritis. Better change the “effect of icariin” in the methods section to biocompatibility as represented in the results. Furthermore, biocompatibility study of these scaffolds on rabbit bone cells would have been appropriate. Please rewrite some of the discussions regarding FTIR and ALP as they are repeated with the results.

Review form: Reviewer 4

Is the manuscript scientifically sound in its present form?

Yes

Are the interpretations and conclusions justified by the results?

Yes

Is the language acceptable?

Yes

Is it clear how to access all supporting data?

Not Applicable

Do you have any ethical concerns with this paper?

No

Have you any concerns about statistical analyses in this paper?

No

Recommendation?

Major revision is needed (please make suggestions in comments)

Comments to the Author(s)

In this manuscript, the author investigated the preparation of PLGA/icariin scaffold by electrospinning and evaluation of it *in vitro* and *in vivo*. This is interesting manuscript. However the authors should address the following issues:

1. In page 4 line 41, what is the state of the word “rough”. I could not observe the difference between PLGA and PLGA/icariin samples in Fig. 1.

2. In page 4 line 46, is the sentence "the hydrophilic materials ..." correct? You should list references in order to insist on this phenomenon. In general, the contact angle of the surface where protein is most adsorbed is about 70-90°.
3. In Fig. 2 caption, (D) is missing.
4. Fig. 4 shows only absorbance. You should evaluate the amount of released icariin from scaffold. By determining the correlation between the concentration of icariin and the absorbance, the correlation between the amount of elution in PBS and time can be clarified. In addition, these data is needed for your suggestion. (page 6 line 43)
5. Fig. 6 shows the adhesion of chondrocyte 4 days after seeding. As you mentioned in Fig. 3, the cells generally proliferate in 3 days. I could not understand why you chose 4 days after seeding. If you want to refer to cell adhesion, initial adhesion of cells should be observed. And the result of initial adhesion would relate the result of proliferation assay in Fig. 5.
6. Fig. 7 shows the ALP activity. There is no information for the cell number of MC3T3-E1. Usually ALP activity is obtained by dividing by DNA or the number of cells. This is because if the number of cells is large, total ALP activity also increase. You should investigate the relation between cell number and ALP activity.
7. In Fig. 8, 9,10, I could not find the information which scaffold (0.01, 0.1%) were used for in vivo assay.
8. In Fig. 8B, white balance should be done.
9. Fig. 9 shows the results of PCR. In Fig. 9, PLGA/icariin, PLGA are arranged in this order. However other figures are in reverse order. It leads to confusion.
10. In gene expression analysis, the expression increases exponentially. Therefore, when making a relative comparison, it is usual to compare the difference between the expression of samples in the scale of 100 and 1000 times. The result of Fig. 9 compared very small differences. You should check whether the analysis method is correct.
11. In my opinion, if you want to mention the effect of sustained release of icariin from scaffold, you should use the PLGA scaffold adding icariin containing medium in vitro, or injecting icariin solution in vivo as a control.

Decision letter (RSOS-181159.R0)

27-Sep-2018

Dear Dr Wang:

Manuscript ID RSOS-181159 entitled "PLGA scaffold loaded with icariin for inhibiting the progression of osteoarthritis in rabbits" which you submitted to Royal Society Open Science, has been reviewed. The comments from reviewers are included at the bottom of this letter.

In view of the criticisms of the reviewers, the manuscript has been rejected in its current form. However, a new manuscript may be submitted which takes into consideration these comments.

Please note that resubmitting your manuscript does not guarantee eventual acceptance, and that your resubmission will be subject to peer review before a decision is made.

Once you have revised your manuscript, go to <https://mc.manuscriptcentral.com/rsos> and login to your Author Center. Click on "Manuscripts with Decisions," and then click on "Create a

Resubmission" located next to the manuscript number. Then, follow the steps for resubmitting your manuscript.

Your resubmitted manuscript should be submitted by 27-Mar-2019. If you are unable to submit by this date please contact the Editorial Office.

Please note that Royal Society Open Science will introduce article processing charges for all new submissions received from 1 January 2018. Charges will also apply to papers transferred to Royal Society Open Science from other Royal Society Publishing journals, as well as papers submitted as part of our collaboration with the Royal Society of Chemistry (<http://rsos.royalsocietypublishing.org/chemistry>). If your manuscript is submitted and accepted for publication after 1 Jan 2018, you will be asked to pay the article processing charge, unless you request a waiver and this is approved by Royal Society Publishing. You can find out more about the charges at <http://rsos.royalsocietypublishing.org/page/charges>. Should you have any queries, please contact openscience@royalsociety.org.

on behalf of Dr Michael Doube (Associate Editor) and Professor Kevin Padian (Subject Editor)
openscience@royalsociety.org

Subject Editor Comments to Author:

Thank you for your submission. It has been read by four reviewers and our AE. Several reviewers raise substantial (and different) concerns. We will entertain a resubmission of this manuscript, but the authors must address ALL comments by the reviewers AND also present a publishable version of the English. We will send a new manuscript out for review but if the reviewers feel that their comments have not been addressed or the English is not publishable, unfortunately we will not be able to consider it further. Best of luck revising and thanks for submitting with RSOS.

Associate Editor Comments to Author (Dr Michael Doube):

Associate Editor: 1

Comments to the Author:

Dear Dr Wang,

Thank you for sending us your manuscript. Four reviewers have seen your manuscript and they have a number of major suggestions that you will need to address before we can consider a resubmission. I am curious about the interpretations around increased chondrocyte proliferation. In the context of osteoarthritis, chondrone formation (an OA hallmark) appears to be due to too much chondrocyte proliferation. So an OA treatment might not necessarily benefit from stimulating chondrocyte proliferation. Please also pay attention to the English used in your manuscript, which would benefit from being seen by a scientific English editor.

Kind regards,
Michael Doube

Reviewers' Comments to Author:

Reviewer: 1

Comments to the Author(s)

Comments to the authors:

Previous studies have indicated that Icaria, a component extracted from epimedium, enhances synthesis of cartilage extracellular matrix (ECM) *in vivo*. However, it remains largely unclear about possible effect of Icaria on pathological conditions of osteoarthritis (OA). In this paper, the authors found that Icaria was released steadily from PLGA/Icaria scaffold. Next, proliferation of chondrocytes on PLGA/Icaria scaffold, but not PLGA scaffold only, was enhanced. Further, implantation of the material into OA model rabbits resulted in increased amount of cartilage ECM. I think that this manuscript contains potentially interesting observation. However, I'm afraid that the results seem to be low impact (see below). Thus, I cannot recommend the acceptance of this manuscript in Royal Society Open Science.

Major concerns:

1. It has been already reported that Icaria promotes synthesis of cartilage ECM as well as osteogenesis. I cannot find a novelty of result of this study.
2. They found that PLGA/Icaria, but not PLGA only, enhances proliferation of chondrocytes on PLGA scaffold (Fig 5). However, Icaria suppresses chondrocyte differentiation via inhibition of TGF- β signaling (Li YC et al., *Clin Exp Pharmacol Physiol* 2013; 40; 635-43). Thus, it might be inappropriate to use Icaria as therapy of OA. Further, the result didn't show chondrocyte proliferation in an Icaria-dose dependent manner. Unfortunately, the result won't show role of Icaria in chondrocyte proliferation. I suggest you will try this study in lower concentration.

Minor concerns:

1. It would be desirable to study Icaria release in the different concentrations (Fig4). The study would enhance quality of data.
2. They must show pictures of affected tissues stained with HE or Toluidin blue before the material implantation (Fig 8). It would be an important point to study effect of the implantation on phenotypes by comparing between pre- and post-implantation.
3. There're many typos through the whole manuscript. 'Inhibited' should be 'inhibit' (line 2 in page 2). 'The express' should be 'the expression' (line 43 in page 5). 'In previous study show' should be 'A previous study showed' (line 5 in page 7). As you know, a word described at first time in the manuscript should be formal name. In this case, OPN should be Osteopontin (line 18, page 7). Transcripts should be as italic (line 21 in page 7). 'diseases affects' should be 'disease that affects' (line 35 in page 7). 'the progress' should be 'the progression' (line 42 in page 7). 'the data of' should be 'the data by' (line 43 in page 7). The fonts are different (ref no 28 and 36). Pages are incomplete (ref no 7, 8, 15, 36, 40). Title should be indicated as upright type (ref no 29).

Reviewer: 2

Comments to the Author(s)

In this manuscript, icaria was released by the poly (lactic-co-glycolic acid (PLGA) electrospun delivery system for prevention of osteoarthritis. Icaria was released slowly and steadily from the scaffold in addition to enhanced chondrocyte proliferation and adhesion. Authors showed sufficient evidence that the icaria has improved the osteogenesis in addition to the previous work shown by using similar electrospun polymer networks.

Reviewer: 3

Comments to the Author(s)

The manuscript from Wang et al. entitled "PLGA scaffold loaded with icariin for inhibiting the progression of osteoarthritis in rabbits" presents the effect of Icariin from PLGA scaffolds on inhibiting the progression of osteoarthritis. The article mainly demonstrated the feasibility of PLGA/Icariin scaffolds for tissue engineering, which was tested by investigating cell biocompatibility *in vitro*, and the therapeutic effects on osteoarthritis *in vivo*. The article was concise and well written. The article could be published after addressing these comments. The introduction should be improved with sufficient comparisons relevant to other technologies such as supercritical fluid technology and 3D printing, which have been utilized for the generation of scaffolds and delivery of active substances for cell growth. I suggest citing these articles relevant to these technologies. *Adv. Healthcare Mater.* 2017, 1700433, *International Journal of Nanomedicine* 2018:13 4227–4245, *J. of Supercritical Fluids* 120 (2017) 43–51, *International Journal of Nanomedicine* 2017:12 1877–1890, *Biofabrication* 9 (2017) 032002, *ACS Biomater. Sci. Eng.* 2018, 4, 800–818, *Materials* 2018, 11, 1390, *Polymers* 2018, 10, 807, *Nanomaterials* 2018, 8, 360.

Better input the respective wavenumber in the graph (Fig. 3) rather than in the figure legend. Loading amount of Icariin?, please represent the release amount in the form of percentage in Figure 4.

I do not think chondrocyte proliferation concerning that showing the effect of Icariin is necessary, as the aim is towards inhibiting the progression of arthritis. Better change the "effect of icariin" in the methods section to biocompatibility as represented in the results. Furthermore, biocompatibility study of these scaffolds on rabbit bone cells would have been appropriate. Please rewrite some of the discussions regarding FTIR and ALP as they are repeated with the results.

Reviewer: 4

Comments to the Author(s)

In this manuscript, the author investigated the preparation of PLGA/icariin scaffold by electrospinning and evaluation of it *in vitro* and *in vivo*. This is interesting manuscript. However the authors should address the following issues:

1. In page 4 line 41, what is the state of the word "rough". I could not observe the difference between PLGA and PLGA/icariin samples in Fig. 1.
2. In page 4 line 46, is the sentence "the hydrophilic materials ..." correct? You should list references in order to insist on this phenomenon. In general, the contact angle of the surface where protein is most adsorbed is about 70-90°.
3. In Fig. 2 caption, (D) is missing.
4. Fig. 4 shows only absorbance. You should evaluate the amount of released icariin from scaffold. By determining the correlation between the concentration of icariin and the absorbance, the correlation between the amount of elution in PBS and time can be clarified. In addition, these data is needed for your suggestion. (page 6 line 43)
5. Fig. 6 shows the adhesion of chondrocyte 4 days after seeding. As you mentioned in Fig. 3, the cells generally proliferate in 3 days. I could not understand why you chose 4 days after seeding. If you want to refer to cell adhesion, initial adhesion of cells should be observed. And the result of initial adhesion would relate the result of proliferation assay in Fig. 5.
6. Fig. 7 shows the ALP activity. There is no information for the cell number of MC3T3-E1. Usually ALP activity is obtained by dividing by DNA or the number of cells. This is because if the number of cells is large, total ALP activity also increase. You should investigate the relation between cell number and ALP activity.

7. In Fig. 8, 9,10, I could not find the information which scaffold (0.01, 0.1 1%) were used for in vivo assay.
8. In Fig. 8B, white balance should be done.
9. Fig. 9 shows the results of PCR. In Fig. 9, PLGA/icariin, PLGA are arranged in this order. However other figures are in reverse order. It leads to confusion.
10. In gene expression analysis, the expression increases exponentially. Therefore, when making a relative comparison, it is usual to compare the difference between the expression of samples in the scale of 100 and 1000 times. The result of Fig. 9 compared very small differences. You should check whether the analysis method is correct.
11. In my opinion, if you want to mention the effect of sustained release of icariin from scaffold, you should use the PLGA scaffold adding icariin containing medium in vitro, or injecting icariin solution in vivo as a control.

Author's Response to Decision Letter for (RSOS-181159.R0)

See Appendix A.

RSOS-181877.R0

Review form: Reviewer 1

Is the manuscript scientifically sound in its present form?

Yes

Are the interpretations and conclusions justified by the results?

Yes

Is the language acceptable?

Yes

Is it clear how to access all supporting data?

Yes

Do you have any ethical concerns with this paper?

No

Have you any concerns about statistical analyses in this paper?

No

Recommendation?

Accept as is

Comments to the Author(s)

They responded point by point to many concerns we pointed out, and further modified the whole manuscript. We appreciate the manuscript is much improved, and agree to the contents of the

manuscript. Thus, we'd like to recommend the acceptance of this manuscript in Royal Society Open Science. They must respond to minor concerns that preclude the acceptance of this manuscript in the journal (see below).

Ref no 7 lacks pages.

Review form: Reviewer 3

Is the manuscript scientifically sound in its present form?

Yes

Are the interpretations and conclusions justified by the results?

Yes

Is the language acceptable?

Yes

Is it clear how to access all supporting data?

Not Applicable

Do you have any ethical concerns with this paper?

No

Have you any concerns about statistical analyses in this paper?

No

Recommendation?

Accept as is

Comments to the Author(s)

Accept

Review form: Reviewer 4

Is the manuscript scientifically sound in its present form?

Yes

Are the interpretations and conclusions justified by the results?

Yes

Is the language acceptable?

Yes

Is it clear how to access all supporting data?

Not Applicable

Do you have any ethical concerns with this paper?

No

Have you any concerns about statistical analyses in this paper?

No

Recommendation?

Accept with minor revision (please list in comments)

Comments to the Author(s)

Authors responded well to reviewer's comment. Although the manuscript was improved, the authors should address the following comments:

The authors insisted that the hydrophilic materials are better for protein adsorption and cell adhesion and can retain bioactivity for the long-term. In general, it is known that the protein adsorption and cell adhesion are highest when the contact angle is around 80°. Your results showed that adding icariin lowered the contact angle to 67°, but you cannot insist that these results led to improve protein adsorption and cell adhesion. If there is a prior research that PLGA is different from general materials on contact angle and cell adhesion, please show the references.

Decision letter (RSOS-181877.R0)

04-Jan-2019

Dear Dr Wang

On behalf of the Editor, I am pleased to inform you that your Manuscript RSOS-181877 entitled "PLGA scaffold carrying icariin to inhibit the progression of osteoarthritis in rabbits" has been accepted for publication in Royal Society Open Science subject to minor revision in accordance with the referee suggestions. Please find the referees' comments at the end of this email.

The reviewers and Subject Editor have recommended publication, but also suggest some minor revisions to your manuscript. Therefore, I invite you to respond to the comments and revise your manuscript.

- Ethics statement

- Data accessibility

It is a condition of publication that all supporting data are made available either as supplementary information or preferably in a suitable permanent repository. The data accessibility section should state where the article's supporting data can be accessed. This section should also include details, where possible of where to access other relevant research materials such as statistical tools, protocols, software etc can be accessed. If the data has been deposited in an external repository this section should list the database, accession number and link to the DOI for all data from the article that has been made publicly available. Data sets that have been

deposited in an external repository and have a DOI should also be appropriately cited in the manuscript and included in the reference list.

If you wish to submit your supporting data or code to Dryad (<http://datadryad.org/>), or modify your current submission to dryad, please use the following link:
<http://datadryad.org/submit?journalID=RSOS&manu=RSOS-181877>

- **Competing interests**

- **Authors' contributions**

- **Acknowledgements**

- **Funding statement**

Because the schedule for publication is very tight, it is a condition of publication that you submit the revised version of your manuscript before 13-Jan-2019. Please note that the revision deadline will expire at 00.00am on this date. If you do not think you will be able to meet this date please let me know immediately.

When submitting your revised manuscript, you will be able to respond to the comments made by the referees and upload a file "Response to Referees" in "Section 6 - File Upload". You can use this

to document any changes you make to the original manuscript. In order to expedite the processing of the revised manuscript, please be as specific as possible in your response to the referees.

Please note that Royal Society Open Science charge article processing charges for all new submissions that are accepted for publication. Charges will also apply to papers transferred to Royal Society Open Science from other Royal Society Publishing journals, as well as papers submitted as part of our collaboration with the Royal Society of Chemistry (<http://rsos.royalsocietypublishing.org/chemistry>). If your manuscript is newly submitted and subsequently accepted for publication, you will be asked to pay the article processing charge, unless you request a waiver and this is approved by Royal Society Publishing. You can find out more about the charges at <http://rsos.royalsocietypublishing.org/page/charges>. Should you have any queries, please contact openscience@royalsociety.org.

on behalf of Dr Michael Doube (Associate Editor) and Kevin Padian (Subject Editor)
openscience@royalsociety.org

Associate Editor Comments to Author (Dr Michael Doube):

Thank you for attending to the reviewers' comments. One reviewer commented that:

" Your results showed that adding icariin lowered the contact angle to 67 °, but you cannot insist that these results led to improved protein adsorption and cell adhesion. If there is a prior research that PLGA is different from general materials on contact angle and cell adhesion, please show the references. "

Please take care to correct your manuscript in line with this advice.

In addition, I have some comments that will aid the clarity of the text:

'better'. In general it's preferred to avoid this word because it is non-specific and value-loaded. In all places where you have used the word 'better' in your manuscript, please replace it with something more specific: higher, faster, more complete, etc.

P1L33 needs a preposition, e.g. *_by_* electrospinning, *_during_* electrospinning, etc.

P1L44 absorption -> resorption

P1L54 relieve pain -> pain relief

P1L56 Traditional ... kidney. Rephrase into correct English.

P1L57 Herbs ... kidney ... Korea. This is a problematic sentence. In what manner do herbs 'nourish' the kidney? Bear in mind that RSOS is a science journal not an alternative medicine publication. Please give the Latin binomial name for the plant or plants from which epimedium is derived. By all means introduce some cultural context about traditional practice but any statements of cause and effect must be backed up by solid empirical evidence. You could write something like, "Chinese medicine practitioners believe that there is a strong link between bones and the kidneys, and that herbs such as epimedium (<insert the Latin Binomial name>) improve bone health by nourishing the kidneys. Icariin is the main pharmacologically active molecule present in epimedium (EP)."

P2L5 believed -> found (a belief by itself is not enough, we need to indicate that there was empirical evidence)

P2L16 four centuries. I find it unlikely that people have been electrospinning nanofibre scaffolds since the 1500s. The reference you cite suggests in the abstract (unfortunately I don't have the whole article) that the technology underlying electrospinning goes back 4 centuries. But not electrospinning itself. Rephrase this sentence.

P3L22 quality -> do you mean quantity?

P4L5 medicine -> Medicine (proper noun)

P4L10 anaesthetization -> anaesthetic induction

P4L29 (and elsewhere) micro-CT -> X-ray microtomography (XMT)

P5L5 smooth -> reduce in roughness

P5L13 Hydrophilic ... long term -> Provide evidence for this statement or remove it.

Results: remove references to methods in your results. You restate your method in the first sentence of each results section - delete these restatements after making sure that they are included in the methods section. Also, rather than "As shown in Fig 8A ... scaffolds", write the sentence directly: "The expression ... scaffolds (Fig 8A)." Please do that for every results section.

P6L46 hyperplastic bones -> do you mean osteophytosis? Please clarify.

P6L54 absorption -> resorption

Figures - scale bars are too small and low contrast. Please make them visible. XMT images and dissection images lack scale bars - put some in.

Reviewer comments to Author:

Reviewer: 3

Comments to the Author(s)

Accept

Reviewer: 1

Comments to the Author(s)

They responded point by point to many concerns we pointed out, and further modified the whole manuscript. We appreciate the manuscript is much improved, and agree to the contents of the manuscript. Thus, we'd like to recommend the acceptance of this manuscript in Royal Society Open Science. They must respond to minor concerns that preclude the acceptance of this manuscript in the journal (see below).

Ref no 7 lacks pages.

Reviewer: 4

Comments to the Author(s)

Authors responded well to reviewer's comment. Although the manuscript was improved, the authors should address the following comments:

The authors insisted that the hydrophilic materials are better for protein adsorption and cell adhesion and can retain bioactivity for the long-term. In general, it is known that the protein adsorption and cell adhesion are highest when the contact angle is around 80°. Your results showed that adding icariin lowered the contact angle to 67°, but you cannot insist that these results led to improve protein adsorption and cell adhesion. If there is a prior research that PLGA is different from general materials on contact angle and cell adhesion, please show the references.

Author's Response to Decision Letter for (RSOS-181877.R0)

See Appendix B.

Decision letter (RSOS-181877.R1)

15-Jan-2019

Dear Dr Wang,

I am pleased to inform you that your manuscript entitled "PLGA scaffold carrying icariin to inhibit the progression of osteoarthritis in rabbits" is now accepted for publication in Royal Society Open Science.

on behalf of Dr Michael Doube (Associate Editor) and Kevin Padian (Subject Editor)
openscience@royalsociety.org

Appendix A

Dear editors,

Thank the editors and reviewers for these precious comments concerning my manuscript entitled “PLGA scaffold carrying icariin to inhibit the progression of osteoarthritis in rabbits”. These comments are all valuable and very helpful for revising and improving my paper, as well as the important guiding significance to my researches. We have studied comments carefully and have made corrections which we hope meet with approval. The responds to the reviewer`s comments are as follow:

Associate Editor: 1

Thank you for sending us your manuscript. Four reviewers have seen your manuscript and they have a number of major suggestions that you will need to address before we can consider a resubmission. I am curious about the interpretations around increased chondrocyte proliferation. In the context of osteoarthritis, chondrone formation (an OA hallmark) appears to be due to too much chondrocyte proliferation. So an OA treatment might not necessarily benefit from stimulating chondrocyte proliferation. Please also pay attention to the English used in your manuscript, which would benefit from being seen by a scientific English editor.

We are appreciative of the editor`s suggestion. The gathering and increase of chondrocytes is the characteristics of OA. In my opinion, like cough and vomiting, it is the response of self-healing or self-protecting in our body, not the cause of OA.

We are very sorry for the mistakes in this manuscript and inconvenience they caused in your reading. The manuscript has been thoroughly revised and edited by the highly qualified native English speaking editors at American Journal Experts (Certificate Verification Key: 1F3A-576F-4B36-D454-8166), so we hope it can meet the journal`s standard. Thanks so much for your useful comments.

Reviewer #1:

Previous studies have indicated that Icariin, a component extracted from epimedium, enhances synthesis of cartilage extracellular matrix (ECM) in vivo. However, it remains largely unclear about possible effect of Icariin on pathological conditions of osteoarthritis (OA). In this paper, the authors

found that Icariin was released steadily from PLGA/Icariin scaffold. Next, proliferation of chondrocytes on PLGA/Icariin scaffold, but not PLGA scaffold only, was enhanced. Further, implantation of the material into OA model rabbits resulted in increased amount of cartilage ECM. I think that this manuscript contains potentially interesting observation. However, I'm afraid that the results seem to be low impact (see below). Thus, I cannot recommend the acceptance of this manuscript in Royal Society Open Science.

Major concerns:

1. It has been already reported that Icariin promotes synthesis of cartilage ECM as well as osteogenesis. I cannot find a novelty of result of this study.

As for the referee's concern, we have indicated the novelty of result of this study. First of all, Icariin is a small molecular weight (only 676.65) component. When it was added into the hydrogel directly, Icariin will release fast into the surrounding tissue fluid. The fast release leads to high carrying dose requirement and short duration of validity, and even more risk of side effects because of the high initial released concentration. (T. Yuan, et al. Conjugated icariin promotes tissue-engineered cartilage formation in hyaluronic acid/collagen hydrogel. DOI: [10.1016/j.procbio.2015.09.006](https://doi.org/10.1016/j.procbio.2015.09.006)). In this study, icariin was loading in the PLGA scaffolds which could maintain a lasting release of icariin. Secondly, this study found that the therapeutic effect of icariin on osteoarthritis depends not only on promoting synthesis of cartilage ECM, but also on inhibiting the absorption of subchondral bone. Thirdly, although Icariin can't induce the chondrogenic differentiate of MC3T3-E1 in vitro, it can promote the synthesis of ECM in vivo. Therefore, we predicted that there is a different mechanism of icariin in vitro and in vivo.

2. They found that PLGA/Icariin, but not PLGA only, enhances proliferation of chondrocytes on PLGA scaffold (Fig 5). However, Icariin suppresses chondrocyte differentiation via inhibition of TGF- β signaling (Li YC et al., *Clin Exp Pharmacol Physiol* 2013; 40; 635-43). Thus, it might be inappropriate to use Icariin as therapy of OA. Further, the result didn't show chondrocyte proliferation in an Icariin-dose dependent manner. Unfortunately, the result won't show role of Icariin in chondrocyte proliferation. I suggest you will try this study in lower concentration.

We are appreciative of the reviewer's suggestion. Li YC et al. verified that Icariin suppresses chondrocyte differentiation via inhibition of TGF- β signaling.

The result is consistent with our data (Real-time polymerase chain reaction assay of articular cartilage). As shown in Fig. 10D, at the concentration of 0.01% and 0.1%, icariin led to the down-regulation the expression of OPN. OPN was synthesized by bone-forming cells and hypertrophic chondrocytes, and previous study suggested that OPN mRNA was found in cartilage from patients with osteoarthritis, while no OPN mRNA expression was observed in chondrocytes of adult healthy cartilage (C. M. Giachelli, Steitz, S. 2000 Osteopontin: a versatile regulator of inflammation and biomineralization. *Matrix. Biol.* . **19**, 615-622.// O. Pullig, G. Weseloh, S. Gauer, Swoboda, B. 2000 Osteopontin is expressed by adult human osteoarthritic chondrocytes: protein and mRNA analysis of normal and osteoarthritic cartilage. *Matrix. Biol.* **19**, 245-255). The gathering and increase of chondrocytes is the characteristics of OA. In my opinion, like cough and vomiting, it is the response of self-healing or self-protecting in our body, not the cause of OA.

We added a MTT assay using a pure icariin solution. Our data showed that the cells cultured with icariin solution showed chondrocyte proliferation in an Icariin-dose dependent manner. At icariin concentrations of about 0.3%, the absorbance decreased, which may due to the cytotoxicity of icariin. However, the result didn't show chondrocyte proliferation in PLGA/Icariin group. The result indicates that the slow release of icariin from scaffolds.

Minor concerns:

1. It would be desirable to study Icariin release in the different concentrations (Fig4). The study would enhance quality of data.

As for the referee's concern, an release experiment in the different concentrations (0.01%, 0.1%, and 1%) has been supplemented.

2. They must show pictures of affected tissues stained with HE or Toluidin blue before the material implantation (Fig 8). It would be an important point to study effect of the implantation on phenotypes by comparing between pre- and post-implantation.

As for the referee's concern, we have added pre-implantation and post-implantation pictures in Fig 8.

3. There're many typos through the whole manuscript. 'Inhibited' should be 'inhibit' (lane 2 in page 2). 'The express' should be 'the expression' (line 43 in page 5). 'In previous study show' should be 'A previous study showed' (line 5 in page 7). As you know, a word described at first time in the

manuscript should be formal name. In this case, OPN should be Osteopontin (line 18, page 7). Transcripts should be as italic (line 21 in page 7). ‘diseases affects’ should be ‘disease that affects’ (line 35 in page 7). ‘the progress’ should be ‘the progression’ (line 42 in page 7). ‘the data of’ should be ‘the data by’ (line 43 in page 7). The fonts are different (ref no 28 and 36). Pages are incomplete (ref no 7, 8, 15, 36, 40). Title should be indicated as upright type (ref no 29).

We are very sorry for the mistakes in this manuscript and inconvenience they caused in your reading. The manuscript has been thoroughly revised and edited by the highly qualified native English speaking editors at American Journal Experts (Certificate Verification Key: 1F3A-576F-4B36-D454-8166), so we hope it can meet the journal's standard. Thanks so much for your useful comments.

Reviewer #2:

In this manuscript, icariin was released by the poly (lactic-co-glycolic acid (PLGA) electrospun delivery system for prevention of osteoarthritis. Icariin was released slowly and steadily from the scaffold in addition to enhanced chondrocyte proliferation and adhesion. Authors showed sufficient evidence that the icariin has improved the osteogenesis in addition to the previous work shown by using similar electrospun polymer networks.

Thank you for your comments.

Reviewer #3:

The manuscript from Wang et al. entitled “PLGA scaffold loaded with icariin for inhibiting the progression of osteoarthritis in rabbits” presents the effect of Icariin from PLGA scaffolds on inhibiting the progression of osteoarthritis. The article mainly demonstrated the feasibility of PLGA/Icariin scaffolds for tissue engineering, which was tested by investigating cell biocompatibility in vitro, and the therapeutic effects on osteoarthritis in vivo. The article was concise and well written. The article could be published after addressing these comments.

1. The introduction should be improved with sufficient comparisons relevant to other technologies such as supercritical fluid technology and 3D printing, which have been utilized for the generation of scaffolds and delivery of active substances for cell growth. I suggest citing these articles

relevant to these technologies. *Adv. Healthcare Mater.* 2017, 1700433, *International Journal of Nanomedicine* 2018:13 4227–4245, *J. of Supercritical Fluids* 120 (2017) 43–51, *International Journal of Nanomedicine* 2017:12 1877–1890, *Biofabrication* 9 (2017) 032002, *ACS Biomater. Sci. Eng.* 2018, 4, 800–818, *Materials* 2018, 11, 1390, *Polymers* 2018, 10, 807, *Nanomaterials* 2018, 8, 360.

We are appreciative of the reviewer's suggestion. Indeed, it will be more profound if we add comparisons relevant to other technologies. We mention in the limitations of the study that we will improve the scaffold to facilitate implantation, and load the cytokines that promote differentiation into chondrocytes. However, the limitation of technical condition leads to the use of electrospinning.

2. Better input the respective wavenumber in the graph (Fig. 3) rather than in the figure legend. Loading amount of Icariin?,

As for the referee's concern, we have added the respective wavenumbers to the Figure.

3. Please represent the release amount in the form of percentage in Figure 4.

As for the referee's concern, we have made change in Figure 4.

4. I do not think chondrocyte proliferation concerning that showing the effect of Icariin is necessary, as the aim is towards inhibiting the progression of arthritis. Better change the "effect of icariin" in the methods section to biocompatibility as represented in the results. Furthermore, biocompatibility study of these scaffolds on rabbit bone cells would have been appropriate. Please rewrite some of the discussions regarding FTIR and ALP as they are repeated with the results.

Considering the reviewer's suggestion, we have made change in the manuscript. And we have rewritten the discussions regarding FTIR and ALP.

Reviewer #4:

In this manuscript, the author investigated the preparation of PLGA/icariin scaffold by electrospinning and evaluation of it in vitro and in vivo. This is interesting manuscript.

However the authors should address the following issues:

1. In page 4 line 41, what is the state of the word “rough”. I could not observe the difference between PLGA and PLGA/icariin samples in Fig. 1.

As for the referee`s concern, scanning electron microscope has been supplemented. And we have made change in the manuscript.

2. In page 4 line 46, is the sentence “the hydrophilic materials ...” correct? You should list references in order to insist on this phenomenon. In general, the contact angle of the surface where protein is most adsorbed is about 70-90°.

We are very sorry for the mistakes in this manuscript and inconvenience they caused in your reading. The manuscript has been thoroughly revised and edited by the highly qualified native English speaking editors at American Journal Experts (Certificate Verification Key: 1F3A-576F-4B36-D454-8166), so we hope it can meet the journal`s standard. Thanks so much for your useful comments.

The contact angle of the material can be affected by surface free energy, the surface smoothness, chemical inhomogeneity, etc. A previous study shows that neat PLGA film shows a contact angle of 75°. With the increase of Ag content, the contact angle increases from 75° for PLGA pristine polymer, to 76° for PLGA/1Ag, to 87° for PLGA/3Ag, and to 91° for PLGA/7Ag (Mariangela S. et al. Antimicrobial Properties and Cytocompatibility of PLGA/Ag Nanocomposites. DOI: 10.3390/ma9010037). Another study shows that water contact angles of PLGA porous scaffold is $95.6 \pm 6.2^\circ$, and that of PLGA/Hydroxyapatite porous scaffold is $84.6 \pm 7.5^\circ$ (Jun Z. et al. Improving osteogenesis of PLGA/HA porous scaffolds based on dual delivery of BMP-2 and IGF-1 via a polydopamine coating. DOI: 10.1039/c7ra12062a).

3. In Fig. 2 caption, (D) is missing.

As for the referee`s concern, we have added the caption to the Figure.

4. Fig. 4 shows only absorbance. You should evaluate the amount of released icariin from scaffold. By determining the correlation between the concentration of icariin and the absorbance, the correlation between the amount of elution in PBS and time can be clarified. In addition, these data is needed for your suggestion. (page 6 line 43)

As for the referee`s concern, an release experiment in the different

concentrations (0.01%, 0.1%, and 1%) has been supplemented. And the amount of released icariin from scaffold is evaluated. Based on our data the release profile of icariin showed that there were burst releases from the PLGA/Icariin composite scaffolds within 12 hours in three groups, and the release subsequently slowed down and maintained icariin release until 192 hours. In the PLGA/0.01% icariin group, 67.5% of the loaded drug had been released after 12 hours, while 7.2% of the loaded drug had been released in the PLGA/0.1% icariin group. The PLGA/0.1% icariin group showed a sustained and slow release.

5. Fig. 6 shows the adhesion of chondrocyte 4 days after seeding. As you mentioned in Fig. 3, the cells generally proliferate in 3 days. I could not understand why you chose 4 days after seeding. If you want to refer to cell adhesion, initial adhesion of cells should be observed. And the result of initial adhesion would relate the result of proliferation assay in Fig. 5.

We are very sorry for our mistaken. It should be 24 hours after seeding. We have made correction according to the Reviewer`s comments.

6. Fig. 7 shows the ALP activity. There is no information for the cell number of MC3T3-E1. Usually ALP activity is obtained by dividing by DNA or the number of cells. This is because if the number of cells is large, total ALP activity also increase. You should investigate the relation between cell number and ALP activity.

It is really true as Reviewer suggested that we should investigate the relation between cell number and ALP activity. Measurements were normalized by the number of cells from BCA protein assay. And we have made change in the manuscript.

7. In Fig. 8, 9,10, I could not find the information which scaffold (0.01, 0.1 1%) were used for in vivo assay.

It is really true as reviewer suggested that there is not information of the scaffold. We used the PLGA scaffold loading 0.1% Icariin. We have made changes in the manuscript.

8. In Fig. 8B, white balance should be done.

As for the referee`s concern, we have adjusted the white balance of the picture.

9. Fig. 9 shows the results of PCR. In Fig. 9, PLGA/icariin, PLGA are arranged in this order. However other figures are in reverse order. It leads to confusion.

As for the referee`s concern, we have adjusted the order of PLGA/Icariin and PLGA.

10. In gene expression analysis, the expression increases exponentially. Therefore, when making a relative comparison, it is usual to compare the difference between the expression of samples in the scale of 100 and 1000 times. The result of Fig. 9 compared very small differences. You should check whether the analysis method is correct.

According to previous study, the relative gene expression levels were normalized to GAPDH (Jun Z. et al. Improving osteogenesis of PLGA/HA porous scaffolds based on dual delivery of BMP-2 and IGF-1 via a polydopamine coating. DOI: 10.1039/c7ra12062a). In the present study, the relative gene expressions were normalized by the expression of (GAPDH).

11. In my opinion, if you want to mention the effect of sustained release of icariin from scaffold, you should use the PLGA scaffold adding icariin containing medium in vitro, or injecting icariin solution in vivo as a control.

It is really true as Reviewer suggested that we should use the PLGA scaffold adding icariin containing medium in vitro, or injecting icariin solution in vivo as a control. However, Icariin is a small molecular weight (only 676.65) component. When it was added into the hydrogel directly, Icariin will release fast into the surrounding tissue fluid. The fast release leads to high carrying dose requirement and short duration of validity, and even more risk of side effects because of the high initial released concentration (T. Yuan, et al. Conjugated icariin promotes tissue-engineered cartilage formation in hyaluronic acid/collagen hydrogel. DOI: 10.1016/j.procbio.2015.09.006). That is the reason we design the PLGA/Icariin scaffold. Considering the Reviewer`s suggestion, we added a MTT assay using a pure icariin solution in vitro. Our data showed that the cells cultured with icariin solution showed chondrocyte proliferation in an Icariin-dose dependent manner. At icariin concentrations of about 0.3%, the absorbance decreased, which may due to the cytotoxicity of icariin. However, the result didn`t show chondrocyte proliferation in PLGA/Icariin group. The result indicates that the slow release of icariin from scaffolds.

Appendix B

Dear Dr Michael Doube and Kevin Padian,

I am glad to hear from you. My manuscript is entitled "PLGA scaffold carrying icariin to inhibit the progression of osteoarthritis in rabbits". Follow your advice, I have made changes in the main document and marked it in red. We have studied comments carefully and have made corrections which we hope meet with approval. The responds to the reviewer`s comments are as follow:

Associate Editor:

Please take care to correct your manuscript in line with this advice. In addition, I have some comments that will aid the clarity of the text.

We are appreciative of the editor`s suggestion. I have made changes in the main document and marked it in red.

Reviewer #1:

Ref no 7 lacks pages.

We are very sorry for the mistakes in this manuscript. I have made changes in the main document.

Reviewer #4:

The authors insisted that the hydrophilic materials are better for protein adsorption and cell adhesion and can retain bioactivity for the long-term. In general, it is known that the protein adsorption and cell adhesion are highest when the contact angle is around 80 °. Your results showed that adding icariin lowered the contact angle to 67 °, but you cannot insist that these results led to improve protein adsorption and cell adhesion. If there is a prior research that PLGA is different from general materials on contact angle and cell adhesion, please show the references.

We are very sorry for the mistakes in this manuscript. Our data can`t prove that adding icariin can improve protein adsorption. I have made changes in the main document. However, the results showed that the cell spreading areas cultured with PLGA/Icariin were wider than those cultured with PLGA. Previous study

proved that the hydrophilic surface property and the RGD peptides can specifically facilitate cellular behaviors, including cell adhesion and proliferation [Shin YC et al. Biomimetic Hybrid Nanofiber Sheets Composed of RGD Peptide-Decorated PLGA as Cell-Adhesive Substrates. DOI: 10.3390/jfb6020367]. the proliferation rate of cells can be improved by the materials when the water contact angle is around 20 °[Jun Z et al. Improving osteogenesis of PLGA/HA porous scaffolds based on dual delivery of BMP-2 and IGF-1 via a polydopamine coating. DOI: 10.1039/c7ra12062a. Zhang Y. A Novel Approach via Surface Modification of Degradable Polymers With Adhesive DOPA-IGF-1 for Neural Tissue Engineering. DOI: 10.1016/j.xphs.2018.10.008.]